# Offline Model-Based Optimization via Normalized Maximum Likelihood Estimation

**Justin Fu & Sergey Levine**
Department of Electrical Engineering and Computer Science
University of California, Berkeley
`{justinjfu,svlevine}@eecs.berkeley.edu`

## Abstract

In this work we consider data-driven optimization problems where one must maximize a function given only queries at a fixed set of points. This problem setting emerges in many domains where function evaluation is a complex and expensive process, such as in the design of materials, vehicles, or neural network architectures. Because the available data typically only covers a small manifold of the possible space of inputs, a principal challenge is to be able to construct algorithms that can reason about uncertainty and out-of-distribution values, since a naive optimizer can easily exploit an estimated model to return adversarial inputs. We propose to tackle this problem by leveraging the normalized maximum-likelihood (NML) estimator, which provides a principled approach to handling uncertainty and out-of-distribution inputs. While in the standard formulation NML is intractable, we propose a tractable approximation that allows us to scale our method to high-capacity neural network models. We demonstrate that our method can effectively optimize high-dimensional design problems in a variety of disciplines such as chemistry, biology, and materials engineering.

## 1 Introduction

Many real-world optimization problems involve function evaluations that are the result of expensive or time-consuming process. Examples occur in the design of materials (Mansouri Tehrani et al., 2018), proteins (Brookes et al., 2019; Kumar & Levine, 2019), neural network architectures (Zoph & Le, 2016), or vehicles (Hoburg & Abbeel, 2014). Rather than settling for a slow and expensive optimization process through repeated function evaluations, one may instead adopt a data-driven approach, where a large dataset of previously collected input-output pairs is given in lieu of running expensive function queries. Not only could this approach be more economical, but in some domains, such as in the design of drugs or vehicles, function evaluations pose safety concerns and an online method may simply be impractical. We refer to this setting as the *offline model-based optimization* (MBO) problem, where a static dataset is available but function queries are not allowed.

A straightforward method to solving offline MBO problems would be to estimate a proxy of the ground truth function $\hat{f}_\theta$ using supervised learning, and to optimize the input $x$ with respect to this proxy. However, this approach is brittle and prone to failure, because the model-fitting process often has little control over the values of the proxy function on inputs outside of the training set. An algorithm that directly optimizes $\hat{f}_\theta$ could easily exploit the proxy to produce adversarial inputs that nevertheless are scored highly under $\hat{f}_\theta$ (Kumar & Levine, 2019; Fannjiang & Listgarten, 2020).

In order to counteract the effects of model exploitation, we propose to use the normalized maximum likelihood framework (NML) (Barron et al., 1998). The NML estimator produces the distribution closest to the MLE assuming an *adversarial* output label, and has been shown to be effective for resisting adversarial attacks (Bibas et al., 2019). Moreover, NML provides a principled approach to generating uncertainty estimates which allows it to reason about out-of-distribution queries. However, because NML is typically intractable except for a handful of special cases (Roos et al., 2008), we show in this work we can circumvent intractability issues with NML in order to construct a reliable and robust method for MBO. Because of its general formulation, the NML distribution pro-

vides a flexible approach to constructing conservative and robust estimators using high-dimensional models such as neural networks.

The main contribution of this work is to develop an offline MBO algorithm that utilizes a novel approximation to the NML distribution to obtain an uncertainty-aware forward model for optimization, which we call NEMO (**N**ormalized maximum likelihood **E**stimation for **M**odel-based **O**ptimization). The basic premise of NEMO is to construct a conditional NML distribution that maps inputs to a distribution over outputs. While constructing the NML distribution is intractable in general, we discuss novel methods to amortize the computational cost of NML, which allows us the scale our method to practical problems with high dimensional inputs using neural networks. A separate optimization algorithm can then be used to optimize over the output to any desired confidence level. Theoretically, we provide insight into why NML is useful for the MBO setting by showing a regret bound for modeling the ground truth function. Empirically, we evaluate our method on a selection of tasks from the Design Benchmark (Anonymous, 2021), where we show that our method performs competitively with state-of-the-art baselines. Additionally, we provide a qualitative analysis of the uncertainty estimates produced by NEMO, showing that it provides reasonable uncertainty estimates, while commonly used methods such as ensembles can produce erroneous estimates that are both confident and wrong in low-data regimes.

## 2 RELATED WORK

**Derivative-free optimization** methods are typically used in settings where only function evaluations are available. This includes methods such as REINFORCE (Williams, 1992) and reward-weighted regression (Peters & Schaal, 2007) in reinforcement learning, the cross-entropy method (Rubinstein, 1999), latent variable models (Garnelo et al., 2018; Kim et al., 2019), and Bayesian optimization (Snoek et al., 2012; Shahriari et al., 2015). Of these approaches, Bayesian optimization is the most often used when function evaluations are expensive and limited. However, all of the aforementioned methods focus on the active or online setting, whereas in this work, we are concerned with the offline setting where additional function evaluations are not available.

**Normalized maximum likelihood** is an information-theoretic framework based on the minimum description length principle (Rissanen, 1978). While the standard NML formulation is purely generative, the conditional or predictive NML setting can be used Rissanen & Roos (2007); Fogel & Feder (2018) for supervised learning and prediction problems. Bibas et al. (2019) apply this framework for prediction using deep neural networks, but require an expensive finetuning process for every input. The goal of our work is to provide a scalable and tractable method to approximate the CNML distribution, and we apply this framework to offline optimization problems.

Like CNML, **conformal prediction** (Shafer & Vovk, 2008) is concerned with predicting the value of a query point $\hat{y}_{t+1}$ given a prior dataset, and provides per-instance confidence intervals, based on how consistent the new input is with the rest of the dataset. Our work instead relies on the NML framework, where the NML regret serves a similar purpose for measuring how close a new query point is to existing, known data.

**The offline model-based optimization** problem has been applied to problems such as designing DNA (Killoran et al., 2017), drugs (Popova et al., 2018), or materials (Hautier et al., 2010). The estimation of distribution algorithm (Bengoetxea et al., 2001) alternates between searching in the input space and model space using a maximum likelihood objective. Kumar & Levine (2019) propose to learn an inverse mapping from output values to input values, and optimize over the output values which produce consistent input values. Brookes et al. (2019) propose CbAS, which uses a trust-region to limit exploitation of the model. Fannjiang & Listgarten (2020) casts the MBO problem as a minimax game based on the *oracle gap*, or the value between the ground truth function and the estimated function. In contrast to these works, we develop an approach to MBO which explicitly reasons about uncertainty. Approaches which utilize uncertainty, such as Bayesian optimization, are commonly used in online settings, and we expect these to work in offline settings as well.

There are several related areas that could arguably be viewed as special cases of MBO. One is in contextual bandits under the batch learning from bandit feedback setting, where learning is often done on logged experience (Swaminathan & Joachims, 2015; Joachims et al., 2018), or offline reinforcement learning (Levine et al., 2020), where model-based methods construct estimates of the MDP

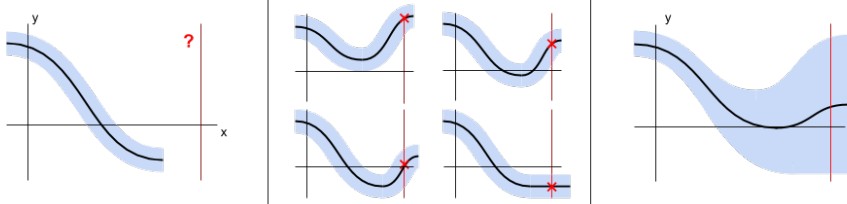

Figure 1: An illustrative example of the construction of the CNML distribution. **Left** We wish to estimate the $p(y|x)$ at some query point $x$, marked by the vertical red line. **Middle** We compute the MLE assuming we knew the true label $y$, for every possible value of $y$. **Right** Finally, we normalize the predictions across all MLE models to produce $p_{\text{NML}}$. The final prediction will likely exhibit large amounts of uncertainty on queries $x$ far from the dataset, because it is easier for the individual MLE estimates to overfit to these outliers.

parameters (Kidambi et al., 2020; Yu et al., 2020). Our work focuses on a more generic function optimization setting, but could be applied in these domains as well.

## 3 PRELIMINARIES

We begin by reviewing the problem formulation for offline model-based optimization, as well as necessary background on the normalized maximum likelihood estimator.

**Problem statement**. We define the offline model-based optimization (MBO) problem as follows. Assume the existence of a stochastic ground truth function $f(y|\boldsymbol{x})$. The MBO algorithm is given a dataset $\mathcal{D}$ of inputs $\boldsymbol{x}$ along with outputs $y$ sampled from $f(y|\boldsymbol{x})$. Like in standard optimization problems, the goal of MBO is to find the input value that maximizes the true function:

$$\boldsymbol{x}^* = \text{argmax}_{\boldsymbol{x}} \mathbb{E}_{y \sim f(y|\boldsymbol{x})}[y]. \tag{1}$$

However, in offline MBO, the algorithm is not allowed to query the true function $f(y|\boldsymbol{x})$, and must find the best possible point $\boldsymbol{x}^*$ using only the guidance of a fixed dataset $\mathcal{D} = \{\boldsymbol{x}_{1:N}, y_{1:N}\}$. One approach to solving this problem is to introduce a separate proxy function $\hat{f}_\theta(y|\boldsymbol{x}) \approx f(y|\boldsymbol{x})$, which is learned from $\mathcal{D}$ as an estimate of the true function. From here, standard optimization algorithms such as gradient descent can be used to find the optimum of the proxy function, $\hat{\boldsymbol{x}}^* = \text{argmax}_{\boldsymbol{x}} \mathbb{E}_{y \sim \hat{f}_\theta(y|\boldsymbol{x})}[y]$. Alternatively, a trivial algorithm could be to select the highest-performing point in the dataset. While adversarial ground truth functions can easily be constructed where this is the best one can do (e.g., if $f(x) = -\infty$ on any $x \notin \mathcal{D}$), in many reasonable domains it should be possible to perform better than the best point in the dataset.

**Conditional normalized maximum likelihood**. In order to produce a conditional distribution $p_{\text{NML}}(y|\boldsymbol{x})$ we can use for estimating the ground truth function, we leverage the conditional or predictive NML (CNML) framework (Rissanen & Roos, 2007; Fogel & Feder, 2018; Bibas et al., 2019). Intuitively, the CNML distribution is the distribution closest to the MLE assuming the test label $y$ is chosen adversarially. This is useful for the MBO setting since we do not know the ground truth value $y$ at points we are querying during optimization, and the CNML distribution gives us conservative estimates that help mitigate model exploitation (see Fig. 1). Formally, the CNML estimator is the minimax solution to a notion of regret, called the *individual regret* defined as $\text{Regret}_{\text{ind}}(h, y) = \log p(y|\boldsymbol{x}, \hat{\theta}_{\mathcal{D} \cup (\boldsymbol{x}, y)}) - \log h(y|\boldsymbol{x})$, and $p_{\text{NML}}(y|\boldsymbol{x}) = \arg\min_h \max_{y'} \text{Regret}_{\text{ind}}(h, y')$ (Fogel & Feder, 2018). The notation $\mathcal{D} \cup (\boldsymbol{x}, y)$ refers to an augmented dataset by appending a query point and label $(\boldsymbol{x}, y)$, to a fixed offline dataset $\mathcal{D}$, and $\hat{\theta}_{\mathcal{D} \cup (\boldsymbol{x}, y)}$ denotes the MLE estimate for this augmented dataset. The query point $(\boldsymbol{x}, y)$ serves to represent the test point we are interested in modeling. The solution to the minimax problem can be expressed as (Fogel & Feder, 2018):

$$p_{\text{NML}}(y|\boldsymbol{x}) = \frac{p(y|\boldsymbol{x}, \hat{\theta}_{\mathcal{D} \cup (\boldsymbol{x}, y)})}{\int_{y'} p(y'|\boldsymbol{x}, \hat{\theta}_{\mathcal{D} \cup (\boldsymbol{x}, y')}) dy'}, \tag{2}$$

where $\hat{\theta}_{\mathcal{D} \cup (\boldsymbol{x}, y)} = \arg\max_\theta \frac{1}{N+1} \sum_{(\boldsymbol{x}, y) \in \mathcal{D} \cup (\boldsymbol{x}, y)} \log p(y|\boldsymbol{x}, \theta)$ is the maximum likelihood estimate for $p$ using the dataset $\mathcal{D}$ augmented with $(\boldsymbol{x}, y)$.

---

**Algorithm 1** NEMO: Normalized Maximum Likelihood for Model-Based Optimization

---

**Input** Model class $\{f_\theta : \theta \in \Theta\}$, Dataset $\mathcal{D} = (\boldsymbol{x}_{1:N}, y_{1:N})$, number of bins $K$, evaluation function $g(y)$, learning rates $\alpha_\theta, \alpha_x$.
**Initialize** $K$ models $\theta_0^{1:K}$, optimization iterate $\boldsymbol{x}_0$
Quantize $y_{1:N}$ into $K$ bins, denoted as $\lfloor \mathcal{Y} \rfloor = \{\lfloor y_1 \rfloor, \cdots \lfloor y_k \rfloor\}$.
**for** iteration $t$ in $1 \ldots T$ **do**
    **for** $k$ in $1 \ldots K$ **do**
        construct augmented dataset: $\mathcal{D}' \leftarrow \mathcal{D} \cup (\boldsymbol{x}_t, \lfloor y_k \rfloor)$.
        update model: $\theta_{t+1}^k \leftarrow \theta_t^k + \alpha_\theta \nabla_{\theta_t^k} \text{LogLikelihood}(\theta_t^k, \mathcal{D}')$
    **end for**
    estimate CNML distribution: $\hat{p}_{\text{NML}}(y|x_t) \propto p(y|\boldsymbol{x}_t, \theta_t^y) / \sum_k p(\lfloor y_k \rfloor | \boldsymbol{x}_t, \theta_t^k)$
    Update $\boldsymbol{x}$: $\boldsymbol{x}_{t+1} \leftarrow \boldsymbol{x}_t + \alpha_x \nabla_x E_{y \sim \hat{p}_{\text{NML}}(y|\boldsymbol{x})}[g(y)]$
**end for**

---

The NML family of estimators has connections to Bayesian methods, and has shown to be asymptotically equivalent to Bayesian inference under the uninformative Jeffreys prior (Rissanen, 1996). NML and Bayesian modeling both suffer from intractability, albeit for different reasons. Bayesian modeling is generally intractable outside of special choices of the prior and model class $\Theta$ where conjugacy can be exploited. On the other hand, NML is intractable because the denominator requires integrating and training a MLE estimator for every possible $y$. One of the primary contributions of this paper is to discuss how to approximate this intractable computation with a tractable one that is sufficient for optimization on challenging problems, which we discuss in Section 4.

## 4 NEMO: NORMALIZED MAXIMUM LIKELIHOOD ESTIMATION FOR MODEL-BASED OPTIMIZATION

We now present NEMO, our proposed algorithm for high-dimensional offline MBO. NEMO is a tractable scheme for estimating and optimizing the estimated expected value of the target function under the CNML distribution. As mentioned above, the CNML estimator (Eqn. 2) is difficult to compute directly, because it requires **a)** obtaining the MLE for each value of $y$, and **b)** integrating these estimates over $y$. In this section, we describe how to address these two issues, using amortization and quantization. We outline the high-level pseudocode in Algorithm 1, and presented a more detailed implementation in Appendix A.2.1.

### 4.1 AN ITERATIVE ALGORITHM FOR MODEL-BASED OPTIMIZATION

We first describe the overall structure of our algorithm, which addresses issue **a)**, the intractability of computing an MLE estimate for every point we wish to query. In this section we assume that the domain of $y$ is discrete, and describe in the following section how we utilize a quantization scheme to approximate a continuous $y$ with a discrete one.

Recall from Section 3 that we wish to construct a proxy for the ground truth, which we will then optimize with gradient ascent. The most straightforward way to integrate NML and MBO would be to fully compute the NML distribution described by Eqn. 2 at each optimization step, conditioned on the current optimization iterate $\boldsymbol{x}_t$. This would produce a conditional distribution $p_{\text{NML}}(y|\boldsymbol{x})$ over output values, and we can optimize $\boldsymbol{x}_t$ with respect to some function of this distribution, such as the mean. While this method is tractable to implement for small problems, it will still be significantly slower than standard optimization methods, because it requires finding the MLE estimate for every $y$ value per iteration of the algorithm. This can easily become prohibitively expensive when using large neural networks on high-dimensional problems.

To remedy this problem, we propose to amortize the learning process by incrementally learning the NML distribution while optimizing the iterate $\boldsymbol{x}_t$. In order to do this, we maintain one model per value of $y$, $\hat{\theta}_k$, each corresponding to one element in the normalizing constant of the NML distribution. During each step of the algorithm, we sample a batch of datapoints, and train each model by appending the current iterate $\boldsymbol{x}_t$ as well as a label $y_{t,k}$ to the batch with a weight $w$ (which is typically set to $w = 1/N$). We then perform a number of gradient step on each model, and use

the resulting models to form an estimate of the NML distribution $p_{\text{NML}}(y_t|\boldsymbol{x}_t)$. We then compute a score from the NML distribution, such as the mean, and perform one step of gradient ascent on $\boldsymbol{x}_t$.

While the incremental algorithm produces only an approximation to the true NML distribution, it brings the computational complexity of the resulting algorithm to just $O(K)$ gradient steps per iteration, rather than solving entire inner-loop optimization problems. This brings the computational cost to be comparable to other baseline methods we evaluated for MBO.

## 4.2  QUANTIZATION AND ARCHITECTURE

The next challenge towards developing a practical NML method is addressing issue **b)**, the intractability of integrating over a continuous $y$. We propose to tackle this issue with quantization and a specialized architecture for modeling the ground truth function.

**Quantization**.  One situation in which the denominator is tractable is when the domain of $y$ is discrete and finite. In such a scenario, we could train $K$ models, where $K$ is the size of the domain, and directly sum over the likelihood estimates to compute the normalizing factor.

In order to turn the NML distribution into a tractable, discrete problem, we quantize all outputs in the dataset by flooring each $y$ value to the nearest bin $\lfloor y_k \rfloor$, with the size of each interval defined as $B = (y_{\text{max}} - y_{\text{min}})/K$. While quantization has potential to induce additional rounding errors to the optimization process, we find in our experiments in Section 5 that using moderate value such as $K = 20$ or $K = 40$ provides both a reasonably accurate solution while not being excessively demanding on computation.

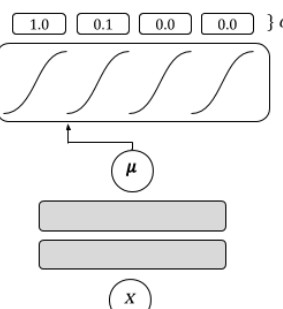

This scheme of quantization can be interpreted as a rectangular quadrature method, where the integral over $y$ is approximated as:

$$\int_y p(y|x, \hat{\theta}_{\mathcal{D} \cup (\boldsymbol{x}, y)}) dy \approx B \sum_{k=1}^{K} p(\lfloor y_k \rfloor | x, \hat{\theta}_{\mathcal{D} \cup (\boldsymbol{x}, \lfloor y_k \rfloor)})$$

Figure 2:  A diagram of the discretized logistic architecture. The mean $\mu$ is denoted by an arrow, which is then passed through offset sigmoid functions to produce $o$.

**Discretized logistic architecture**.  Quantization introduces unique difficulties into the optimization process for MBO. In particular, quantization results in flat regions in the optimization landscape, making using gradient-based algorithms to optimize both inputs $\boldsymbol{x}$ and models $p(y|x, \theta)$ challenging. In order to alleviate these issues, we propose to model the output using a discretized logistic architecture, depicted in Fig. 2. The discretized logistic architecture transforms and input $x$ into the mean parameter of a logistic distribution $\mu(\boldsymbol{x})$, and outputs one minus the CDF of a logistic distribution queried at regular intervals of $1/K$ (recall that the CDF of a logistic distribution is itself the logistic or sigmoid function). Therefore, the final output is a vector $o$ of length K, where element $k$ is equal to $\sigma(\mu(\boldsymbol{x}) + k/K)$. We note that similar architectures have been used elsewhere, such as for modeling the output over pixel intensity values in images (Salimans et al., 2017).

We train this model by first encoding a label $y$ as a vector $y_{disc}$, where $y_{disc}[k \leq \text{bin}(y)] = 1$ and elements $y_{disc}[k > \text{bin}(y)] = 0$. $\text{bin}(y)$ denotes the index of the quantization bin that $y$ falls under. The model is then trained using a standard binary cross entropy loss, applied per-element across the entire output vector. Because the output represents the one minus the CDF, the expected value of the discretized logistic architecture can be computed as $y_{\text{mean}}(\boldsymbol{x}) = \mathbb{E}_{y \sim p(y|x)}[g(y)] = \sum_k [g(k) - g(k-1)]o[k]$. If we assume that $g$ normalizes all output values to $[0, 1]$ in uniform bins after quantization, the mean can easily be computed as a sum over the entire output vector, $y_{\text{mean}} = \frac{1}{K} \sum_k o[k]$

**Optimization**  The benefit of using such an architecture is that when optimizing for $\boldsymbol{x}$, rather than optimizing the predicted output directly, we can compute gradients with respect to the logistic parameter $\mu$. Because $\mu$ is a single scalar output of a feedforward network, it is less susceptible to flat gradients introduced by the quantization procedure. Optimizing with respect to $\mu$ is sensible as it shares the same global optimum as $y_{\text{mean}}$, and gradients with respect to $\mu$ and $y_{\text{mean}}$ share a positive angle, as shown by the following theorem:

**Proposition 4.1** (Discretized Logistic Gradients). *Let $\mu(\boldsymbol{x})$ denote the mean of the discretized logistic architecture for input $\boldsymbol{x}$, and $y_{mean}(\boldsymbol{x})$ denote the predicted mean. Then,*

1. *If $x \in \arg\max_{\boldsymbol{x}} \mu(\boldsymbol{x})$, then $\boldsymbol{x} \in \arg\max_{\boldsymbol{x}} y_{mean}(\boldsymbol{x})$.*

2. *For any $\boldsymbol{x}$, $\langle \nabla_{\boldsymbol{x}} \mu(\boldsymbol{x}), \nabla_{\boldsymbol{x}} y_{mean}(\boldsymbol{x}) \rangle \geq 0$.*

*Proof.* See Appendix. A.1.2. □

### 4.3 THEORETICAL RESULTS

We now highlight some theoretical motivation for using CNML in the MBO setting, and show that estimating the true function with the CNML distribution is close to an expert even if the test label is chosen adversarially, which makes it difficult for an optimizer to exploit the model. As discussed earlier, the CNML distribution minimizes a notion of regret based on the log-loss with respect to an adversarial test distribution. This construction leads the CNML distribution to be very conservative for out-of-distribution inputs. However, the notion of regret in conventional CNML does not easily translate into a statement about the outputs of the function we are optimizing. In order to reconcile these differences, we introduce a new notion of regret, the *functional regret*, which measures the difference between the output estimated under some model against an expert within some function class $\Theta$.

**Definition 4.1** (Functional Regret). *Let $q(y|\boldsymbol{x})$ be an estimated conditional distribution, $\boldsymbol{x}$ represent a query input, and $y^*$ represent a label for $\boldsymbol{x}$. We define the functional regret of a distribution $q$ as:*

$$Regret_f(q, \mathcal{D}, \boldsymbol{x}, y^*) = |\mathbb{E}_{y \sim q(y|\boldsymbol{x})}[g(y)] - \mathbb{E}_{y \sim p(y|\boldsymbol{x}, \hat{\theta})}[g(y)]|$$

*Where $\hat{\theta}$ is MLE estimator for the augmented dataset $\mathcal{D} \cup (\boldsymbol{x}, y^*)$ formed by appending $(\boldsymbol{x}, y^*)$ to $\mathcal{D}$.*

A straightforward choice for the evaluation function $g$ is the identity function $g(y) = y$, in which case the functional regret controls the difference in expected values between $q$ and the MLE estimate $p(y|x, \hat{\theta})$. We now show that the functional regret is bounded for the CNML distribution:

**Theorem 4.1.** *Let $p_{NML}$ be the conditional NML distribution defined in Eqn. 2. Then,*

$$\forall_x \max_{y^*} Regret_f(p_{NML}, \mathcal{D}, x, y^*) \leq 2g_{max}\sqrt{\Gamma(\mathcal{D}, x)/2}.$$

$\Gamma(\mathcal{D}, x) = \log\{\sum_y p(y|x, \hat{\theta}_{\mathcal{D} \cup (\boldsymbol{x}, y)})\}$ *is the minimax individual regret, and $g_{max} = \max_{y \in \mathcal{Y}} g(y)$.*

*Proof.* See Appendix A.1.2. □

This statement states that, for any test input $\boldsymbol{x}$, the CNML estimator is close to the best possible expert if the test label $y$ is chosen adversarially. Importantly, the expert is allowed to see the label of the test point, but the CNML estimator is not, which means that if the true function lies within the model class, this statement effectively controls the discrepancy in performance between the true function and $p_{NML}$. The amount of slack is controlled by the minimax individual regret $\Gamma$ (Fogel & Feder, 2018), which can be interpreted as a measure of uncertainty in the model. For large model classes $\Theta$ and data points $\boldsymbol{x}$ far away from the data, the individual regret is naturally larger as the NML estimator becomes more uncertain, but for data points $\boldsymbol{x}$ close to $\Theta$ the regret becomes very small. This behavior can be easily seen in Fig. 3, where the CNML distribution is very focused in regions close to the data but outputs large uncertainty estimates in out-of-distribution regions.

## 5 EXPERIMENTS

In our experimental evaluation, we aim to **1)** evaluate how well the proposed quantized NML estimator estimates uncertainty in an offline setting, and **2)** compare the performance of NEMO to a number of recently proposed offline MBO algorithms on high-dimensional offline MBO benchmark problems. Our code is available at `https://sites.google.com/view/nemo-anonymous`

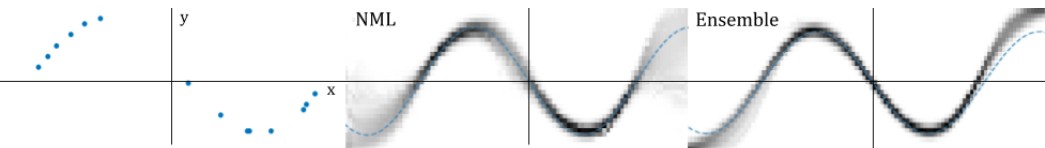

Figure 3: A comparison of uncertainty estimates between quantized NML and bootstrapped ensembles. **Left**: A small training dataset collected using the ground truth function $\sin(x)$ and quantized into 32 bins. Each dot represents a data point. **Middle**: Predictions from quantized CNML, where darker regions indicate outputs with high estimated probability. The ground truth is marked by a dotted blue line. Note that in regions where there is little data, the NML distribution tends to correctly output a very diffuse distribution with uncertain outputs. **Right**: Predictions from a bootstrap ensemble of 32 models. In out-of-support regions far from the data, the bootstrap ensemble tends to underestimate uncertainty and produce overconfident predictions.

## 5.1 MODELING WITH QUANTIZED NML

We begin with an illustrative example of modeling a function with quantized NML. We compared a learned a quantized NML distribution with a bootstrapped ensemble method (Breiman, 1996) on a simple 1-dimensional problem, shown in Fig. 3. The ensemble method is implemented by training 32 neural networks using the same model class as NML, but with resampled datasets and randomized initializations. Both methods are trained on a discretized output, using a softmax cross-entropy loss function. We see that in areas within the support of the data, the NML distribution is both confident and relatively accurate. However, in regions outside of the support, the quantized NML outputs a highly uncertain estimate. In contrast, the ensemble method, even with bootstrapping and random initializations, tends to produce an ensemble of models that all output similar values. Therefore, in regimes outside of the data, the ensemble still outputs highly confident estimates, even though they may be wrong.

## 5.2 HIGH-DIMENSIONAL MODEL-BASED OPTIMIZATION

We evaluated NEMO on a set of high-dimensional MBO problems. The details for the tasks, baselines, and experimental setup are as follows, and hyperparameter choices with additional implementation details can be found in Appendix A.2.

### 5.2.1 TASKS

We evaluated on 6 tasks from the Design-bench (Anonymous, 2021), modeled after real-world design problems for problems in materials engineering (Hamidieh, 2018), biology (Sarkisyan et al., 2016), and chemistry (Gaulton et al., 2012), and simulated robotics. Because we do not have access to a real physical process for evaluating the material and molecule design tasks, Design-bench follows experimental protocol used in prior work (Brookes et al., 2019; Fannjiang & Listgarten, 2020) which obtains a ground truth evaluation function by training a separate regressor model to evaluate the performance of designs. For the robotics tasks, designs are evaluated using the MuJoCo physics simulator (Todorov et al., 2012).

**Superconductor**. The Superconductor task involves designing a superconducting material that has a high critical temperature. The input is space is an 81-dimensional vector, representing properties such as atomic radius and valence of elements which make up the material. This dataset contains a total of 21,263 superconductoring materials proposed by Hamidieh (2018).

**GFP**. The goal of the green fluorescent protein (GFP) task is to design a protein with high fluorescence, based on work proposed by Sarkisyan et al. (2016). This task requires optimizing a 238-dimensional sequence of discrete variables, with each dimension representing one amino acid in the protein and taking on one of 20 values. We parameterize the input space as logits in order to make this discrete problem amenable to continuous optimization. In total, the dataset consists of 5000 such proteins annotated with fluorescence values.

**MoleculeActivity**. The MoleculeActivity task involves designing the substructure of a molecule that exhibits high activity when tested against a target assay (Gaulton et al., 2012). The input space

is represented by 1024 binary variables parameterized by logits, which corresponds to the Morgan radius 2 substructure fingerprints. This dataset contains a total of 4216 data points.

The final 3 tasks, **HopperController**, **AntMorphology**, and **DKittyMorphology**, involve designing robotic agents. HopperController involves learning the parameters of a 5126-parameter neural network controller for the Hopper-v2 task from OpenAI Gym (Brockman et al., 2016). The Ant and DKitty morphology tasks involve optimizing robot parameters such as size, orientation, and joint positions. AntMorphology has 60 parameters, and DKittyMorphology has 56 parameters.

### 5.2.2 Baselines

In addition to NEMO, we evaluate several baseline methods. A logical alternative to NEMO is a forward ensemble method, since both NEMO and ensemble methods maintain a list of multiple models in order to approximate a distribution over the function value, and ensembles are often used to obtain uncertainty-aware models. We implement an ensemble baseline by training $K$ networks on the task dataset with random initializations and bootstrapping, and then optimizing the mean value of the ensemble with gradient ascent. In our results in Table 1, we label the ensemble as "Ensemble" and a single forward model as "Forward". Additionally, we implement a Bayesian optimization baseline wth Gaussian processes (GP-BO) for the Superconductor, GFP, and MoleculeActivity tasks, where we fit the parameters of a kernel and then optimize the expected improvement according to the posterior. We use an RBF kernel for the Superconductor task, and an inner product kernel for the GFP and MoleculeActivity tasks since they have large, discrete input spaces. Note that the GP baseline has no variance between runs since the resulting method is completely deterministic.

We evaluate 3 state-of-the-art methods for offline MBO: model inversion networks (MINs) (Kumar & Levine, 2019), conditioning by adaptive sampling (CbAS) (Brookes et al., 2019), and autofocused oracles (Fannjiang & Listgarten, 2020). MINs train an inverse mapping from outputs $y$ to inputs $x$, and generate candidate inputs by searching over high values of $y$ and evaluating these on the inverse model. CbAS uses a generative model of $p(x)$ as a trust region to prevent model exploitation, and autofocused oracles expands upon CbAS by iteratively updating the learned proxy function and iterates within a minimax game based on a quantity known as the oracle gap.

### 5.3 Results and Discussion

Our results are shown in Table 1. We follow an evaluation protocol used in prior work for design problems (Brookes et al., 2019; Fannjiang & Listgarten, 2020), where the algorithm proposes a set of candidate designs, and the 100th and 50th percentile of scores are reported. This mimics a real-world scenario in which a batch of designs can be synthesized in parallel, and the highest performing designs are selected for use.

For each experiment, we produced a batch of 128 candidate designs. MINs, CbAS, and autofocused oracles all learn a generative model to produce candidate designs, so we sampled this batch from the corresponding model. Ensemble methods and NEMO do not maintain generative models, so we instead optimized a batch of 128 particles. We report results averaged of 16 random seeds.

NEMO outperforms all methods on the Superconductor task by a very large margin, under both the 100th and 50th percentile metrics, and in the HopperController task under the 100th percentile metric. For the remaining tasks (GFP, MoleculeActivity, AntMorphology, and HopperMorphology), NEMO also produces competitive results in line with the best performing algorithm for each task. These results are promising in that NEMO performs consistently well across all 6 domains evaluated, and indicates a significant number of designs found in the GFP and Superconductor task were *better* than the best performing design in the dataset. In Appendix A.3, we present learning curves for NEMO, as well as an ablation study demonstrating the the beneficial effect of NML compared to direct optimization on a proxy function. Note that unlike the prior methods (MINs, CbAS, Autofocused), NEMO *does not* require training a generative model on the data, only a collection of forward models.

| 100th Perc. | Superconductor | GFP | MoleculeActivity |
|---|---|---|---|
| NEMO (ours) | **127.0 ± 7.292** | 3.359 ± 0.036 | **6.682 ± 0.209** |
| Ensemble | 88.01 ± 10.43 | 2.924 ± 0.039 | 6.525 ± 0.1159 |
| Forward | 89.64 ± 9.201 | 2.894 ± 0.001 | 6.636 ± 0.066 |
| MINs | 80.23 ± 10.67 | 3.315 ± 0.033 | 6.508 ± 0.236 |
| CbAS | 72.17 ± 8.652 | **3.408 ± 0.029** | 6.301 ± 0.131 |
| Autofoc. | 77.07 ± 11.11 | **3.365 ± 0.023** | 6.345 ± 0.141 |
| GP-BO | 89.72 ± 0.000 | 2.894 ± 0.000 | **6.745 ± 0.000** |
| Dataset Max | 73.90 | 3.152 | 6.558 |
| | HopperController | AntMorphology | DKittyMorphology |
| NEMO (ours) | **2130.1 ± 506.9** | **393.7 ± 6.135** | **431.6 ± 47.79** |
| Ensemble | 1877.0 ± 704.2 | - | - |
| Forward | 1050.8 ± 284.5 | **399.9 ± 4.941** | 390.7 ± 49.24 |
| MINs | 746.1 ± 636.8 | 388.5 ± 9.085 | 352.9 ± 38.65 |
| CbAS | 547.1 ± 423.9 | **393.0 ± 3.750** | 369.1 ± 60.65 |
| Autofoc. | 443.8 ± 142.9 | 386.9 ± 10.58 | 376.3 ± 47.47 |
| Dataset Max | 1361.6 | 108.5 | 215.9 |

Table 1: 100th percentile ground truth scores and standard deviations over a batch of 128 designs for each task, averaged across 16 trials.

| 50th Perc. | Superconductor | GFP | MoleculeActivity |
|---|---|---|---|
| NEMO (ours) | 66.41 ± 4.618 | **3.219 ± 0.039** | 5.814 ± 0.092 |
| Ensemble | 48.72 ± 2.637 | 2.910 ± 0.020 | **6.412 ± 0.123** |
| Forward | 54.06 ± 5.060 | 2.894 ± 0.000 | **6.401 ± 0.186** |
| MINs | 37.32 ± 10.50 | 3.135 ± 0.019 | 5.806 ± 0.078 |
| CbAS | 32.21 ± 7.255 | **3.269 ± 0.018** | 5.742 ± 0.123 |
| Autofoc. | 31.57 ± 7.457 | **3.216 ± 0.029** | 5.759 ± 0.158 |
| GP-BO | **72.42 ± 0.000** | 2.894 ± 0.000 | 6.373 ± 0.000 |
| | HopperController | AntMorphology | DKittyMorphology |
| NEMO (ours) | 390.2 ± 43.37 | **326.9 ± 5.229** | 180.8 ± 34.94 |
| Ensemble | 362.5 ± 80.09 | - | - |
| Forward | 185.0 ± 72.88 | 318.0 ± 12.05 | **255.3 ± 6.379** |
| MINs | **520.4 ± 301.5** | 184.8 ± 29.52 | 211.6 ± 13.67 |
| CbAS | 132.5 ± 23.88 | 267.3 ± 16.55 | 203.2 ± 3.580 |
| Autofoc. | 116.4 ± 18.66 | 176.7 ± 59.94 | 199.3 ± 8.909 |

Table 2: 50th percentile ground truth scores and standard deviations over a batch of 128 designs for each task, averaged across 16 trials.

# 6 CONCLUSION

We have presented NEMO (Normalized Maximum Likelihood Estimation for Model-Based Optimization), an algorithm that mitigates model exploitation on MBO problems by constructing a conservative model of the true function. NEMO generates a tractable approximation to the NML distribution to provide conservative objective value estimates for out-of-distribution inputs. Our theoretical analysis also suggests that this approach is an effective way to estimate unknown objective functions outside the training distribution. We evaluated NEMO on a number of design problems in materials science, robotics, biology, and chemistry, where we show that it attains very large improvements on two tasks, while performing competitively with respect to prior methods on the other four.

While we studied offline MBO in this paper, we believe that NML is a promising framework for building uncertainty-aware models which could have numerous applications in a variety of other problem settings, such as in off-policy evaluation, batch learning from bandit feedback, or offline reinforcement learning.

## ACKNOWLEDGEMENTS

We would like to thank Brandon Trabucco and Aviral Kumar for assisting with the implementation and evaluation of baselines and helping with the benchmark. This research was funded by the Office of Naval Research, C3.ai, and computational resources from Amazon Web Services.

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

# A  APPENDIX

## A.1  PROOFS

### A.1.1  PROOF OF PROPOSITION 4.1

This statement directly from monotonicity. Because $y_{\text{mean}}$ is a sum of monotonic functions in $\mu$, $y_{\text{mean}}$ must also be a monotonic function of $\mu$. This implies that the global maxima are the same, and that gradients must point in the same direction since $\langle \nabla_{\boldsymbol{x}}\mu, \nabla_{\boldsymbol{x}} y_{\text{mean}} \rangle = \left\langle \nabla_{\boldsymbol{x}}\mu, \frac{dy_{\text{mean}}}{d\mu} \nabla_{\boldsymbol{x}}\mu \right\rangle = \frac{dy_{\text{mean}}}{d\mu} ||\nabla_{\boldsymbol{x}}\mu||_2^2 \geq 0$.

### A.1.2  PROOF OF THEOREM 4.1

In this section we present the proof for Thm. 4.1, restated below:

$$\forall_x \ \max_{y^*} \text{Regret}_f(p_{\text{NML}}, \mathcal{D}, x, y^*) \leq 2g_{\text{max}} \sqrt{\Gamma(\mathcal{D}, x)/2}$$

There are two lemmas we will use in our proof. First, the difference in expected value between two distributions $p(x)$ and $q(x)$ can be bounded by the total variation distance $TV(p, q)$ and the maximum function value $f_{\text{max}} = \max_x f(x)$:

$$|E_{p(x)}[f(x)] - E_{q(x)}[f(x)]| = |\sum_x [p(x) - q(x)]f(x)|$$
$$\leq f_{\text{max}}|\sum_x [p(x) - q(x)]|$$
$$= f_{\text{max}} 2TV(p(x), q(x))$$

Second, Fogel & Feder (2018) show that the NML distribution obtains the best possible minimax individual regret of

$$\max_y \text{Regret}_{\text{ind}}(p_{\text{NML}}, \mathcal{D}, x, y) = \log\{\sum_y p(y|x, \hat{\theta}_{\mathcal{D} \cup (\boldsymbol{x}, y)})\} \overset{\text{def}}{=} \Gamma(\mathcal{D}, x)$$

Using these two facts, we can show:

$$\text{Regret}_f(p_{\text{NML}}, \mathcal{D}, x, y^*) = |\mathbb{E}_{y \sim p_{\text{NML}}(y|x)}[g(y)] - \mathbb{E}_{y \sim p(y|x, \hat{\theta}_{\mathcal{D} \cup (\boldsymbol{x}, y^*)})}[g(y)]|$$
$$\leq 2g_{\text{max}} TV(p(y|x, \hat{\theta}_{\mathcal{D} \cup (\boldsymbol{x}, y^*)}), p_{\text{NML}}(y|x))$$
$$\leq 2g_{\text{max}} \sqrt{\frac{1}{2} KL(p(y|x, \hat{\theta}_{\mathcal{D} \cup (\boldsymbol{x}, y^*)}), p_{\text{NML}}(y|x))}$$
$$\leq 2g_{\text{max}} \sqrt{\frac{1}{2} \max_q \mathbb{E}_{y \sim q(y|x)}[\log p(y|x, \hat{\theta}_{\mathcal{D} \cup (\boldsymbol{x}, y)}) - \log p_{\text{NML}}(y|x)]}$$
$$= 2g_{\text{max}} \sqrt{\Gamma(\mathcal{D}, x)/2}$$

Where we apply the total variation distance lemma from lines 1 to 2. From lines 2 to 3, we used Pinsker's inequality to bound total variation with KL, and from lines 3 to 4 we used the fact that the maximum regret is always greater than the KL, i.e.

$$KL(p(y|x, \hat{\theta}_{\mathcal{D} \cup (\boldsymbol{x}, y^*)}), p_{\text{NML}}(y|x)) = \mathbb{E}_{y \sim p(y|x, \hat{\theta})}[\log p(y|x, \hat{\theta}_{\mathcal{D} \cup (\boldsymbol{x}, y^*)}) - \log p_{\text{NML}}(y|x)]$$
$$\leq \mathbb{E}_{y \sim p(y|x, \hat{\theta})}[\log p(y|x, \hat{\theta}_{\mathcal{D} \cup (\boldsymbol{x}, y)}) - \log p_{\text{NML}}(y|x)]$$
$$\leq \max_q \mathbb{E}_{y \sim q(y|x)}[\log p(y|x, \hat{\theta}_{\mathcal{D} \cup (\boldsymbol{x}, y)}) - \log p_{\text{NML}}(y|x)]$$

On the final step, we substituted the definition of $\Gamma(\mathcal{D}, x)$ as the individual regret of the NML distribution $p_{\text{NML}}$.

## A.2 EXPERIMENTAL DETAILS

### A.2.1 DETAILED PSEUDOCODE

There are a number of additional details we implemented in order to improve the performance of NEMO for high-dimensional tasks. These include:

- Using the Adam optimizer (Kingma & Ba, 2014) rather than stochastic gradient descent.
- Pretraining the models $\theta$ in order to initialize the procedure with an accurate initial model.
- Optimizing over a batch of $M$ designs, in order to follow previous evaluation protocols.
- Optimizing with $x$ with respect to the internal scores $\mu$ instead of $E_{p_{\text{NML}}}[g(y)]$.
- Using target networks for the NML model, originally proposed in reinforcement learning algorithms, to improve the stability of the method.

We present pseudocode for the practical implementation of NEMO below:

---

**Algorithm 2** NEMO – Practical Instantiation

---

**Input** Model class $\Theta$, Dataset $\mathcal{D} = (\boldsymbol{x}_{1:N}, y_{1:N})$, number of bins $K$, batch size $M$, learning rates $\alpha_\theta, \alpha_x$, target update rate $\tau$
**Initialize** $K$ models $\theta_0^{1:K}$
**Initialize** batch of optimization iterates $\mathcal{B}_0 = \boldsymbol{x}_0^{1:M}$ from the **best** performing $\boldsymbol{x}$ in $\mathcal{D}$.
Pretrain $\theta_0^{1:K}$ using supervised learning on $\mathcal{D}$.
**Initialize** target networks $\bar{\theta}_0^{1:K} \leftarrow \theta_0^{1:K}$.
Quantize $y_{1:N}$ into $K$ bins, denoted as $\lfloor \mathcal{Y} \rfloor = \{\lfloor y_1 \rfloor, \cdots \lfloor y_k \rfloor\}$.
**for** iteration $t$ in $1 \ldots T$ **do**
  **for** $k$ in $1 \ldots K$ **do**
    Sample $\boldsymbol{x}_t'$ from batch $\mathcal{B}_t$.
    Construct Augmented dataset: $\mathcal{D}' \leftarrow \mathcal{D} \cup (\boldsymbol{x}_t', \lfloor y_k \rfloor)$
    Compute gradient $g_\theta \leftarrow \nabla_{\theta_t^y} \text{LogLikelihood}(\theta_t^y, \mathcal{D}')$
    Update model $\theta_t^y$ using Adam and $g_\theta$ with learning rate $\alpha_\theta$.
  **end for**
  Update target networks $\bar{\theta}_{t+1}^y \leftarrow \tau \theta_{t+1}^y + (1 - \tau) \bar{\theta}_t^y$
  **for** $m$ in $1 \ldots M$ **do**
    Compute internal values $\mu^k(\boldsymbol{x}_t^m)$ from target networks $\bar{\theta}_t^y$ for all $k \in 1, \cdots, K$
    Compute gradient $g_x$ for $\boldsymbol{x}_t^m$: $\nabla_x \frac{1}{K} \sum_k \mu^k(\boldsymbol{x}_t^m)$
    Update $\boldsymbol{x}_t^m$ using Adam and gradient $g_x$ with learning rate $\alpha_x$.
  **end for**
**end for**

---

### A.2.2 HYPERPARAMETERS

The following table lists the hyperparameter settings we used for each task. We obtained our hyperparameter settings by performing a grid search across different settings of $\alpha_\theta$, $\alpha_x$, and $\tau$. We used 2-layer neural networks with softplus activations for all experiments. We used a smaller networks for GFP, Hopper, Ant, and DKitty (64-dimensional layers) and a lower discretization $K$ for computational performance reasons, but we did not tune over these parameters.

| | Superconductor | MoleculeActivity | GFP | Hopper | Ant | DKitty |
|---|---|---|---|---|---|---|
| Learning rate $\alpha_\theta$ | 0.05 | | | 0.005 | | |
| Learning rate $\alpha_x$ | 0.1 | | 0.01 | 0.001 | | |
| Network Size | 256,256 | | | 64,64 | | |
| Discretization $K$ | 40 | | | 20 | | |
| Batch size $M$ | 128 | | | | | |
| Target update rate $\tau$ | 0.05 | | | | | |

For baseline methods, please refer to Anonymous (2021) for hyperparameter settings.

### A.3 ABLATION STUDIES

In this section, we present 3 ablation studies. The first is on the effect of NML training, by comparing NEMO to optimizing a pretrained baseline neural network. The second ablation study investigates the architecture choice, comparing the discretized logistic architecture to a standard feedforward neural network. The final ablation study investigates the ratio of model optimization steps to input optimization steps. Each logging iteration in these figures corresponds to 50 loop iterations as depicted in Algorithm A.2.1.

#### A.3.1 EFFECT OF NML

In this section, we present learning curves for NML, as well as an ablation study comparing NML (orange curve) against a forward optimization algorithm without NML (blue curve), labeled "No NML". The "No NML" algorithm is identical to the NML algorithm detailed in Alg. A.2.1, except the NML learning rate $\alpha_\theta$ is set to 0.0. This means that the only model training that happens is done during the pretraining step. For illustrative purposes, we initialize the iterates from the worst-performing scores in the dataset to better visualize improvement, rather than initializing from the best scores which we used in our final reported numbers.

The scores on the Superconductor task are shown in the following figure. Removing NML training makes it very difficult for training on most designs, as shown by the poor performance on the $50^{th}$ percentile metric.

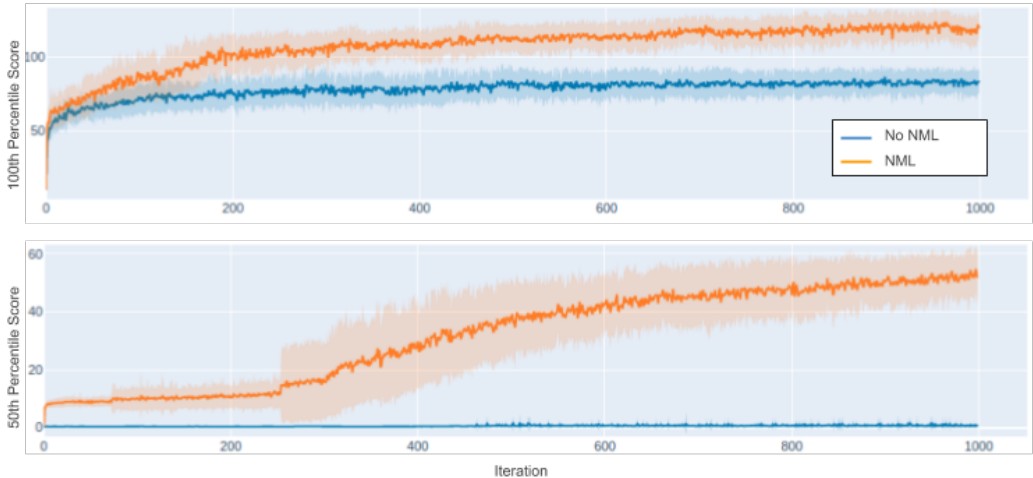

The scores on the MoleculeActivity task follow a similar trend.

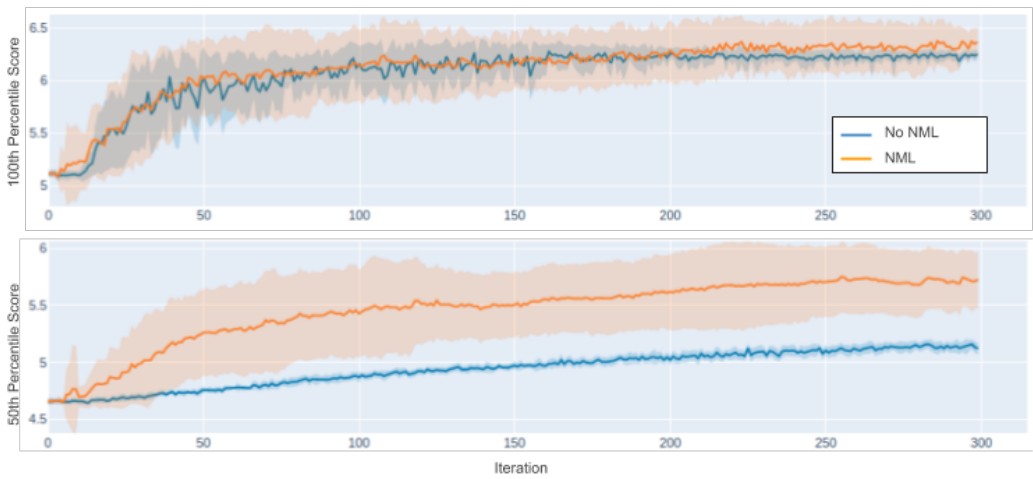

And finally, the scores on the GFP task also display the same trend.

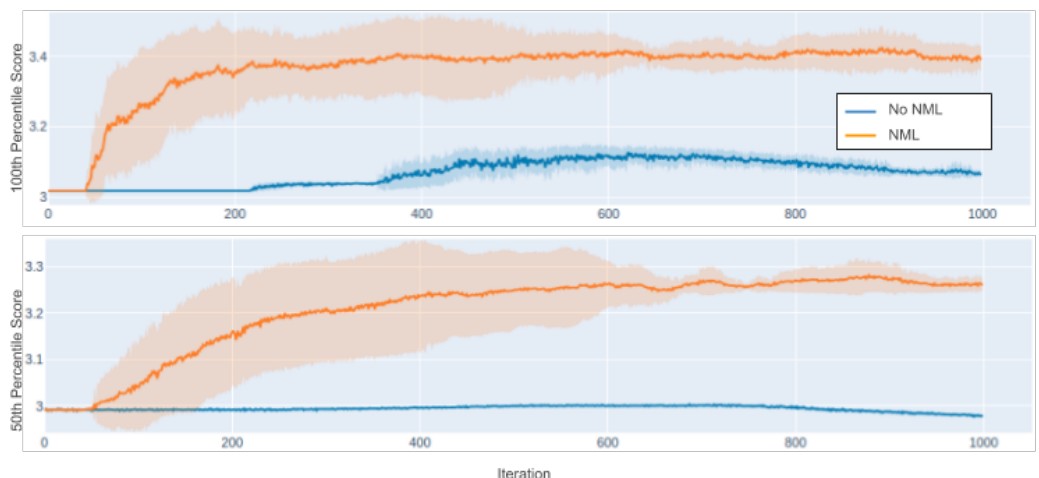

### A.3.2 ARCHITECTURE CHOICE

In this ablation study, we investigate the efficacy of the discretized logistic architecture. As a baseline, we compared against a standard feedforward network, trained with a softmax cross-entropy loss to predict the discretized output $y$. We label this network as "Categorical", because the output of the network is a categorical distribution. All other hyperparameters, including network layer sizes, remain unchanged from those reported in Appendix. A.2.2.

On the Superconductor task, both the discretized logistic and categorical networks score well on the $100^{th}$ percentile metric, but the Categorical architecture displays less consistency in optimizing designs, as given by poor performance on the $50^{th}$ percentile metric.

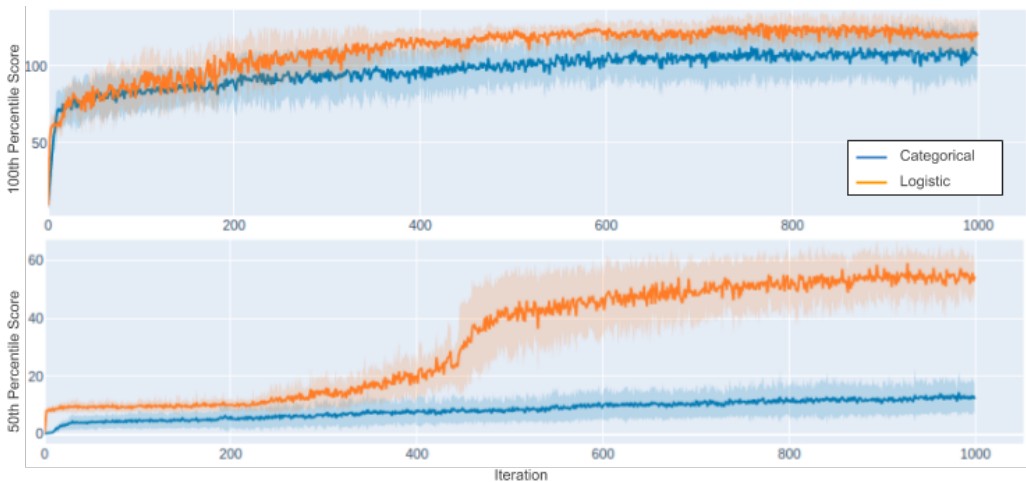

On the MoleculeActivity task, the Categorical network performs comparatively better, but still underperforms the discretized logistic architecture.

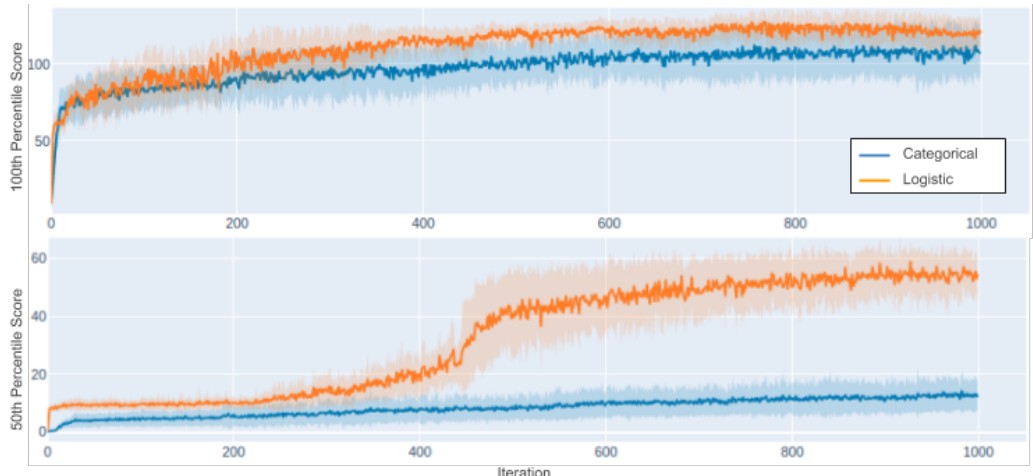

### A.3.3 Ratio of Optimization Steps

In this ablation study, we investigate the effect of the ratio of model optimization steps to input optimization step. For this experiment, we fix the learning rate $\alpha_x$ to the hyperparameter values in Appendix. A.2.2, fix the input optimization steps to 1, and vary the number of model optimization steps we take.

We first investigate the Superconductor task, using a small model learning rate $\alpha_\theta = 0.0005$. In this setting, we see a clear trend that additional model optimization steps are helpful and increase convergence speed.

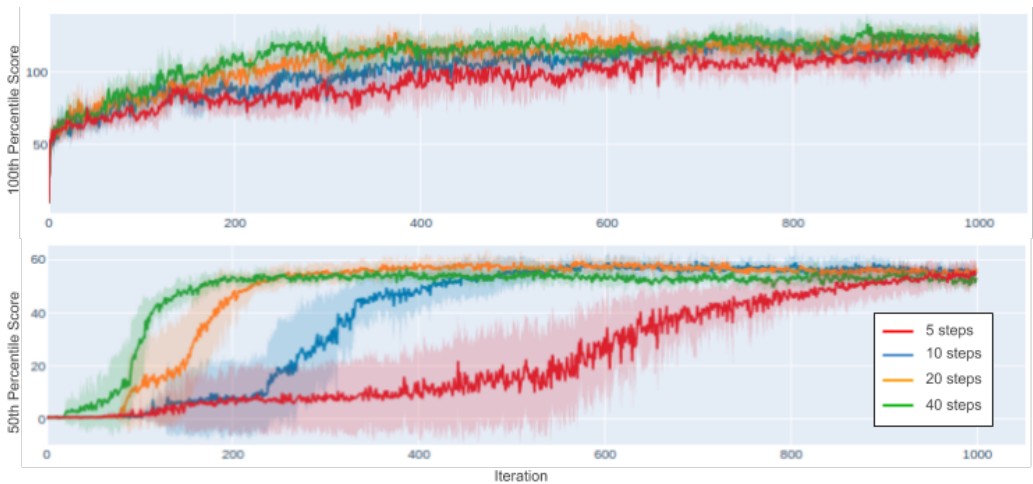

Using a higher learning rate of $\alpha_\theta = 0.05$, a smaller amount of steps works better, which suggests that it is easy to destabilize the learning process using a higher learning rate.

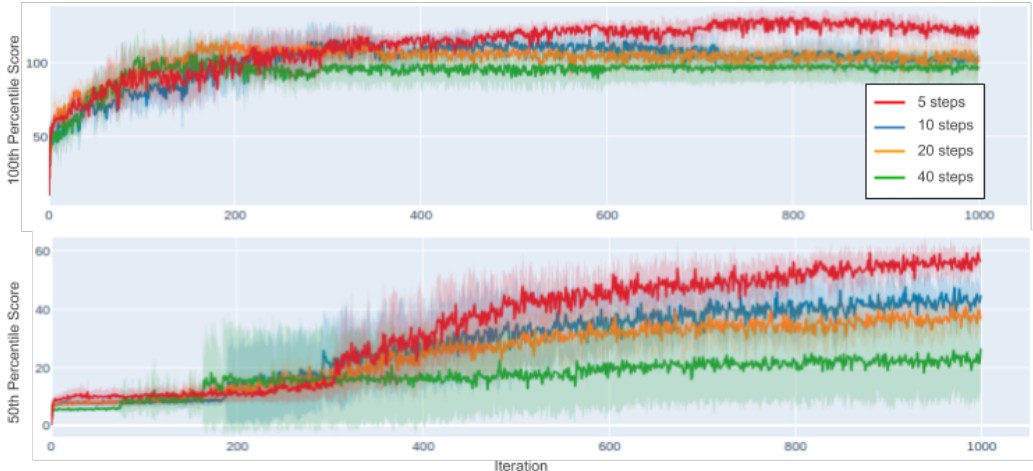

A similar trend holds true in the MoleculeActivity task, albeit less pronounced. The following figure uses a learning rate of $\alpha_\theta = 0.0005$, and we once again see that more model optimization steps leads to increased performance.

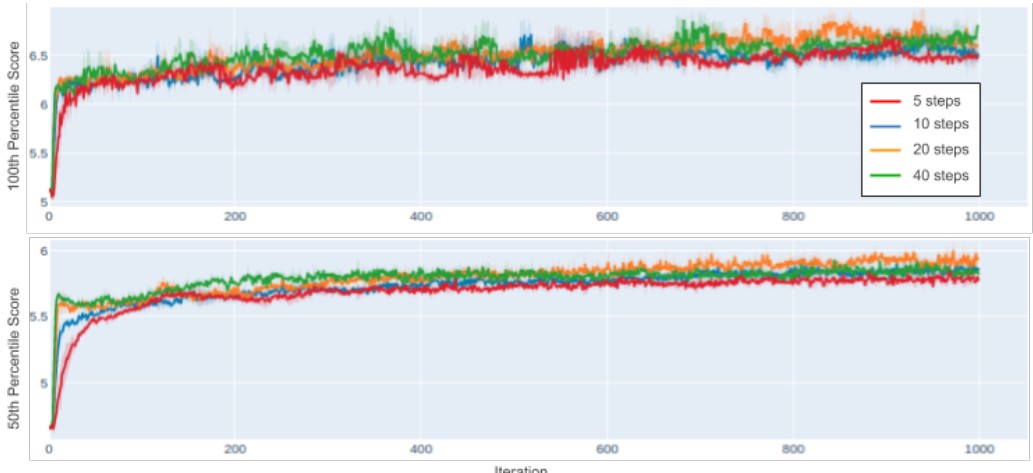

And using a learning rate of $\alpha_\theta = 0.05$, the advantage becomes less clear.

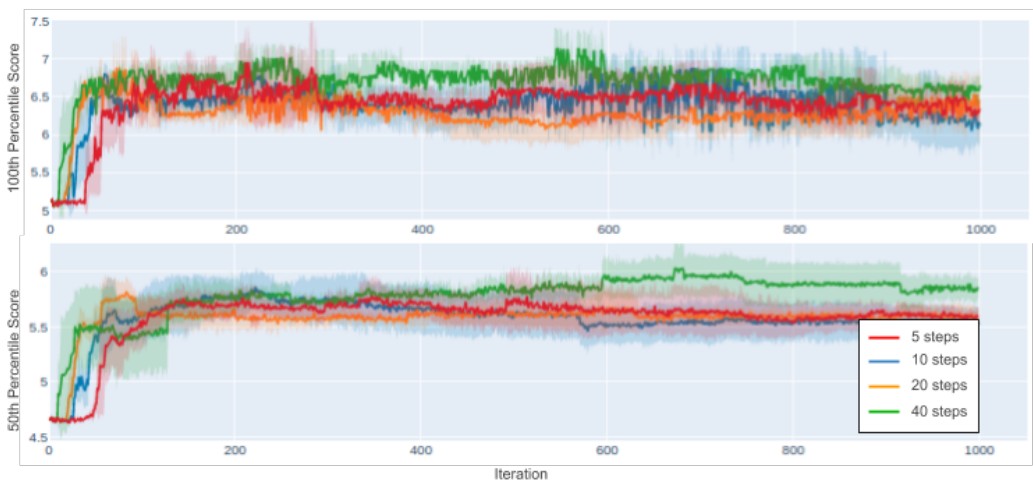

Overall, while we performed a grid search over the learning rates to achieve the highest performance, using a large number of model optimization steps with a small learning rate $\alpha_\theta$ appears to be a consistent strategy which performs well.

