# OpenReview forum: "Offline Model-Based Optimization via Normalized Maximum Likelihood Estimation"
_ICLR.cc/2021/Conference — ICLR 2021 Poster_

### Official Review · AnonReviewer3 · 2020-10-28
**Utilizing normalized maximum-likelihood for better estimation of model uncertainty for offline data-driven optimization**

**Rating:** 6
**Confidence:** 3

**Review:**

Summary: The paper proposes an approximation method, called NEMO (Normalized maximum likelihood Estimation for model-based optimization)  to compute the conditional normalized maximum log-likelihood of a query data point as a way to quantify the uncertainty in a forward prediction model in offline model-based optimization problems. The main idea is to construct a conditional NML (CNML) distribution that maps the high-dimensional inputs to a distribution over output variables. In addition, the paper provides a theoretical motivation that estimating the true function with the CNML is close to the best possible expert even if the test label is chosen adversarially, which is a great challenge for an optimizer to exploit the model. By using this CNML on three offline optimization benchmark datasets (Superconductor, GFP, MoleculeActivity) with gradient ascent-based optimization, the NEMO outputs all the other four baselines on the Superconductor dataset by almost 1.4x to 1.7x, the generate comparable results as the other four baselines method on the GFP and MoleculeActivity datasets.

Typo:
In section 4, “We outline the high-level pseudocode in Algorithm 4” -> “…. in Algorithm 1”

Questions:
1. When sampling a batch of data points at each step of algorithm 1, is the sampling performed with or without replacements?
2. What’s the variance across the 16 random runs? Is the score of the best algorithm in the average performance across 16 random runs significantly different from the baseline algorithms?
3. When estimating the CNML, what is the number of models in the experiments? Are the number of models differ from dataset to dataset? How to choose the number of models in practice?
4. Since the output y needs to be discretized in the NEMO algorithm, how difficult for the algorithm to scale when y is multivariate?

------------------------------------------------------------------------------------------------------------
Update:
I think the authors did a great job of addressing my concerns, I'm happy to increase my score to 6

---

> ### Author Response · Authors · 2020-11-17
> **Author response to reviewer 3**
>
> Thank you for your comments. We have clarified several points below, and please let us know if this addresses your concerns.
>
> > “When sampling a batch of data points at each step of algorithm 1, is the sampling performed with or without replacements?”
>
> Sampling is done with replacement. The model training step is identical to minibatch optimization methods used in standard supervised learning.
>
> > “What’s the variance across the 16 random runs? Is the score of the best algorithm in the average performance across 16 random runs significantly different from the baseline algorithms?”
>
> We have updated Table 1 with standard deviations for all methods. The average score for NEMO is significantly higher than baselines for the Superconductor and newly added HopperController tasks, and is still competitive with state-of-the-art methods for the other tasks.
>
> > “When estimating the CNML, what is the number of models in the experiments? Are the number of models differ from dataset to dataset? How to choose the number of models in practice?”
>
> We use one model per discretization bin. These values are reported in Appendix A.2.2. In practice, we found a value of 20-40 to perform quite well for the problems we tried, and we did not need to tune this hyperparameter.
>
> > “Since the output y needs to be discretized in the NEMO algorithm, how difficult for the algorithm to scale when y is multivariate?”
>
> Because standard optimization problems generally deal with scalar valued objectives, we don’t foresee the need to consider the case when y is multivariate in the context of optimization. However, multi-objective optimization would be a great direction to explore for future work, and it would interesting to see how NML could fit into the analysis of a Pareto frontier.

---

> > ### Author Response · Authors · 2020-11-20
> > **Follow-up**
> >
> > Hello, we were wondering if there were any additional concerns you had, and we would be happy to clarify them. Thank you!

---

### Official Review · AnonReviewer1 · 2020-10-28
**Official Blind Review #1**

**Rating:** 6
**Confidence:** 4

**Review:**

# summary

This paper proposed a method based on NML and provided a principled approach
to estimate the uncertainty for OOD data.


# pros

1.  The method proposed in this work is a principled way to handle uncertainty
    for novel points out of the original dataset compared with, for example,
    deep ensemble.
2.  One clear advantage of ths proposed approach is this method can scale to
    large dataset, compared with GP, which scales cubically.


# cons

1.  The authors claim using a supervised approach is brittle and prone to
    failure as the uncertainty of inputs outside of training data is
    uncontroled. However, this is not true and uncertainty can be well
    controlled depending on the model, which can be non-parametric or
    parametric, distribution-free or distribution-dependent. For example, to
    measure uncertainty on novel point, GP could be viewed as the ground truth
    under some conditions. My question is why not directly compare your
    approach with a GP approach, then combine the posterior with an acquisition
    function, such as EI. The offline MBO problem presented in this work is
    similiar to an online MBO, except we have only one online sample, and
    we are tying to optimize this point. Comparing eq(1) of this paper with the
    formula of EI, it is easy to see eq(1) is exactly EI, if we assume there
    exists (x\*,y\*) such that y\* is larger than objective values in the training
    data set. Thus the problem presented in this work can be effectively solved
    through **one step** of conventional BO. Given the datasets used in the
    experiments of this work is are not of large scale, I think comparing with
    a GP-based BO is necessary.
2.  In Figure 3, although not a major concern, I don't think the comparison
    with the ensemble is fair. Although this work uses bootstrap and ensemble,
    MSE cannot capture uncertainty, thus it is not an ideal metric in this
    setting. For example, to obtain a similar uncertainty estimation compared
    with NML (middle column), we can use a deep ensemble, which optimizes NLL
    instead of MSE.
3.  The experimental results, in my opinion, are not sufficient and there is
    only one table presenting empirical results. I don't want to judge
    sufficiency by the quantity of tables or figures, but considering the
    theoretical analysis is not strong enough, I think more empirical study
    should be performed.
4.  The uncertainty estimation seems too conservative, and this could make the
    estimated uncertainty less useful, especially for high-dimensional
    problems.


# questions

The approach proposed in this paper seems very similiar with conformal
prediction. In conformal prediction, the target value y\* for a novel point x\*
(adversarial input) is chosen so that y\* is compatible with the original
dataset. As I am not familiar with the evaluation protocol in Brookles2019 and
Fannjiang2020, the metric used in Table 1 is not clear to me. Can the authors
say more about that?

update:

---
Overall speaking, the added GP-BO results address my  concerns, and I've updated the score from 5->6. A final update will be given later.

---

> ### Author Response · Authors · 2020-11-17
> **Author response to reviewer 1**
>
> Thank you for the comments and feedback. Our overall impression from your review is that while the method is promising, the empirical study is not thorough enough to back our claims. Therefore, we have significantly expanded our evaluations according to your suggestions, and included a GP-based baseline, additional evaluations on new tasks, and ablation studies for justifying our design choices. Please let us know if this addresses your concerns, or if there are any further modifications you would like us to make.
>
> > “My question is why not directly compare your approach with a GP approach… Thus the problem presented in this work can be effectively solved through one step of conventional BO.”
>
> You are correct in that a Bayesian method such as a GP can be used to accurately measure uncertainty on out-of-distribution inputs in some cases (especially in low-dimensional tasks) and that one step of a method such as BO can be used to solve the offline MBO problem. We have included results for a GP-based method by selecting the design maximizing the expected improvement acquisition function. We use an RBF kernel for the continuous Superconductor task, and a dot-product kernel for the GFP and MoleculeActivity tasks, since they are discrete.
>
> > “The experimental results, in my opinion, are not sufficient and there is only one table presenting empirical results. I don't want to judge sufficiency by the quantity of tables or figures, but considering the theoretical analysis is not strong enough, I think more empirical study should be performed.”
>
> In order to strengthen the empirical analysis, we have included additional experimental results on a 3 robotics domains from the design-bench, where we see similar experimental trends hold true: NEMO consistently ranks among the top performing algorithms and on the HopperController task significantly outperforms prior methods. An additional benefit for these tasks are evaluated in the MuJoCo robotics simulator, and therefore we can be confident the designs are valid.
>
> To justify our design choices, we have also included additional ablation studies on the choice of architecture (thermometer architecture vs standard feedforward architecture) and hyperparameter settings such as learning rates and number of optimization steps, included in Appendix A.3.
>
> > “The approach proposed in this paper seems very similiar with conformal prediction. In conformal prediction, the target value y* for a novel point x* (adversarial input) is chosen so that y* is compatible with the original dataset. “
>
> The connection with conformal prediction is an interesting one we were not aware of. It does seem like conformal prediction could be used in a similar manner to CNML, by providing per-instance confidence intervals that could prevent model exploitation. Additionally, the NML regret may have connections to conformity measures, as they both provide a measure of you close a datapoint is to the rest of the dataset. We think this could be an interesting connection to explore in future work, and could reveal additional theoretical insights on the algorithm. We have added a brief discussion of this to the related work (Section 2).
>
> > “As I am not familiar with the evaluation protocol in Brookles2019 and Fannjiang2020, the metric used in Table 1 is not clear to me. Can the authors say more about that?”
>
> Regarding the metric in Table 1, we report max/median ground truth scores across a batch of 128 candidate designs output by the algorithm. The max score corresponds to what would commonly be done in practice - one would synthesize a batch of candidate designs and use the best performing one. We report the median to verify that most designs perform decently well, because some algorithms may get lucky on scoring a single high-performing design. In terms of physical units, the Superconductor task is reported in Kelvin, the Molecule task in IC50 scores, the GFP in log-fluorescence scores (see Sarkisyan et. al. 2016), and the remaining robotics task in rewards (derived via velocity) from the underlying reinforcement learning MDP.
>
> > “In Figure 3, although not a major concern, I don't think the comparison with the ensemble is fair… For example, to obtain a similar uncertainty estimation compared with NML (middle column), we can use a deep ensemble, which optimizes NLL instead of MSE.”
>
> We are indeed using NLL for both methods to keep the comparison fair. To be more specific, we discretized the output values, and the ensemble method predicts these using a softmax cross-entropy loss. We have updated Section 5.1 to clarify this.

---

> > ### Author Response · Authors · 2020-11-20
> > **Additional concerns?**
> >
> > Again, we appreciate the constructive feedback on the paper. We are wondering if there are any outstanding issues you think should be addressed, or if the current presentation is satisfactory. Thank you.

---

### Official Review · AnonReviewer2 · 2020-10-30
**Reviewer 2**

**Rating:** 8
**Confidence:** 4

**Review:**

Updated review
----
----

# Summary

This work proposes an approach for model-based optimization based on learning a density function through an approximation of the normalized maximum likelihood (NML). This is done by discretizing the space and fitting distinct model parameters for each value. To lower the computational cost, the authors propose optimizing the candidates concurrently with the model parameters. Each model's distribution is encoded as a neural net outputting a scalar which is then encoded using a thermometer approach using a series of shifted sigmoid. Candidates are optimized based on the average value of the scalar of each model evaluated using parameters obtained from an exponentially weighted average of its most recent parameters.

# Reason for score

This work proposes a reasonable approximation to an interesting estimator and demonstrate it is capable of achieving good consistent performance. This is likely to be of interest to the community and, as far as I'm aware, is sufficiently novel. Given that I see no noteworthy issues and all of my major concerns have been addressed, I don't see any reason for rejection. I strongly support acceptance.

# Pros

* Using estimates of the NML for model-based optimization is an interesting idea.
* This work shows that the NML can be successfully approximated with a relatively coarse discretization and that both the optimization of the candidate and the various model parameters can be optimized in tandem. This suggests that this type of approach is viable and possibly warrants further investigation.


Initial review
----
----

# Summary

This work proposes an approach for model-based optimization based on learning a density function through an approximation of the normalized maximum likelihood (NML). This is done by discretizing the space and fitting distinct model parameters for each value. To lower the computational cost, the authors propose optimizing the candidates concurrently with the model parameters. Each model's distribution is encoded as a neural net outputting a scalar which is then encoded using a thermometer approach using a series of shifted sigmoid. Candidates are optimized based on the average value of the scalar of each model evaluated using parameters obtained from an exponentially weighted average of its most recent parameters.

# Reason for score

There are a lot of typos and issues with the notation which make this paper unfit for publication in its current state. Otherwise, the work seems interesting but I find the experiments don't provide much insight. Though the comparison with the selected method is favorable, it's hard to consider them significant when there is a notable difference in only one of the three datasets of an unpublished benchmark. Additionally, the fact that the reported results only cover half the datasets from this benchmark raises some questions. Despite the negative tone of this paragraph, I want to note that the severity of some of these issues is subjective while others can be easily fixed. My mind isn't made up and I hope the authors can clarify anything I might have missed.

# Pros

* Using estimates of the NML for model-based optimization is an interesting idea.
* This work shows that the NML can be successfully approximated with a relatively coarse discretization and that both the optimization of the candidate and the various model parameters can be optimized in tandem. This suggests that this type of approach is viable and possibly warrants further investigation.

# Cons

* The current notation is often confusing and even ambiguous at times. This will probably transpire in some of my other comments, but I do consider these issues to be superficial and easily fixed. I've provided some suggestions below for how to improve the notation. I understand that notation preference is a very subjective thing so the authors should feel free to opt for something different, but I do think the notation needs to be improved.
* The "thermometer encoding" of the output seems like an odd choice, especially given how it is done here. If I understood the approach correctly, this seems to be a hacky way of using the output of the NN, $o_{int}$, as parameters to a logistic distributions. Why not treat it this way directly? My interpretation of this approach is that the $o^k_{int}$s are parameters for logistic distributions and the mean is optimized through the unnormalized probs by optimizing each $o^k_{int}$ directly. Would this be a fair description of the approach? I was expecting the discretization to be used directly to approximate the integral in eq. (2).
* The experimental results aren't very conclusive, only showing a clear benefit in a single case. Without additional results, it is difficult to say much about the behavior and properties of this method. A visualization of the distribution of the value of the candidates might help convey some additional information in favor of this method. It's possible that I am missing some context to appreciate these results. If that is the case, it would help if the authors could provide the context I need to appreciate these results.

# Questions

* I don't think I understand what makes the appendix results an ablation study. From what I understand, these results only compare with the case where there is no learning of the model parameters. What are the models initialized to? Where does the data come into play?
* Have the authors considered using points selected from some fixed quadrature method instead of a uniform grid?
* A common theme for the comparison methods is the idea of not diverging too much from the data. Was the validity of the outputs of this method evaluated in any way? How can I know that the method isn't just exploiting some quirk in the learned models used to evaluate it while some of the other methods avoid doing this?
* Were comparisons with a simple approaches Bayesian optimization tried, e.g., Gaussian process?
* What happens when doing more iterations on the log likelihood before updating $x$? How much do we lose when only doing a single update? Does using a more accurate approximation of the NLM improve/worsen performance? (I was hoping this would be part of the ablation study)
* How do the run times compare?
* Were the other datasets from the Design-bench benchmark tried?

# Misc and typos

* page 2, "in that it has shown to be", missing word?
* page 2, "to discuss how approximate this intractable", missing word?
* page 3, when $p_{NML}$ is formally written, the meaning of $y$ is ambiguous since it is on the LHS and also being redefined by $\max_y$ in the RHS.
* page 3, "The notation $D \cup (x, y)$ refers to an augmented dataset $D \cup (x_{N+1}, y_{N+1})$", this wasn't very informative and felt a bit tautological. I would recommend sticking with one of the two notations.
* page 3, "where $D$ is fixed to the given offline dataset, and $\theta_{D \cup (x,y)}$ [...]", this sentence is a bit confusing as a whole. The start of the sentence talks only about $D$ so the $\theta$ mention is unexpected when reading.
* page 3, right after eq. (2), $(x, y)$ is on the LHS but then also part of the expectation on the RHS. What is the expectation being taken over? Are $x$ and $y$ being redefined?
* page 4, "for y in 1 ... K do", is $y$ I don't believe that $y$ is assumed to be an integer.
* page 4, algorithm 4?
* page 4,  "this would produce a distribution over output values $p(y|x)$", this doesn't "type check" for me. If I understood correctly, $p$ or $p(\bullet | x)$ represent distributions and $y$ are output values. Also, it might be good to reuse the "$\hat p_{NML}$" notation to get the following sentence: "this would produce a conditional distribution, $\hat p_{NML}(\bullet | x_t)$,  over output values.
* page 4, "we can optimize $x_t$ with respect to some evaluation function $g(y)$ of this distribution, such as the
mean", this is confusing since algorithm 1 has $\mathbb{E}[ g(y)]$. What does it mean for the evaluation function $g$ to be the mean in this context? How should I interpret the expectation of mean(y)? Also, I assumed that what was meant is that $g$ is the evaluation function, not $g(y)$, or is $g$ meant to return a function given a $y$?
* page 6, "a straightforward choice for the evaluation function g is the identity function $g(x) = x$", it might be best to stick to a consistent variable name, e.g., $g(y) = y$, to avoid confusion about what the domain of $g$ is.
* Proof of thm 4.1, equation under "using these two facts, we can show:", $q$ should be replaced with $p_{NML}$ in the RHS.
* Proof of thm 4.1, going from TV to KL, looking up the bound returns a $1/2$ factor inside the root rather than a $2$. I could have missed a detail but thought worth mentioning in case I haven't.
* Proof of thm 4.1, missing a "]" when bounding the KL divergence.
* Algorithm 2, there is some undefined superscript $k$ and $y$ in the loop over $x^m_t$.

# Notation suggestions

* When writing expectations, I would strongly recommend making it explicit over which variables they are, e.g., $\mathbb{E}\_{y \sim p(\bullet | x, \theta)}$. Introducing some shorthand notation might help make this more concise, e.g., $p_{x, \theta}(y) := p(y | x, \theta)$.
* When referring to a function, only mention its name/symbol and reserve the form that includes inputs to refer to the output of the function given those inputs, e.g., a function $g$, an evaluation score $g(y)$.
* Avoid relying on the readers pattern matching abilities for assigning meaning to variables and make sure variables, e.g., $\mathbf{x}$ and $y$, are always explicitly defined. By explicitly defined, I include defining $y$ compactly with something like $\max_{y \in \mathcal{Y}} g(y)$, for example. There is not need to be overly verbose but it should never leave room for interpretation. This is related to the point about expectation notation.
* I usually prefer explicit domains, e.g., $\sum_{y \in \mathcal{Y}}$ instead of $\sum_y$, but I consider it fine to omit it if variables names are always reserved to the same domain when reused. This was not the case for $x$ in this paper.
* When writing pseudocode, either loop over integer indices or over elements of a set, it is confusing to use "for y in 1 ... K" when $y$ isn't an integer. Additionally, using both makes it difficult to tell that $k$ is associated with $y$ inside a loop.

---

> ### Author Response · Authors · 2020-11-17
> **Author response to reviewer 2**
>
> Thank you for your detailed feedback. We believe your main concerns are with the depth of empirical study and writing/notational issues in the paper. We have significantly expanded the experimental evaluation with additional ablation studies on the architecture choice and optimization parameters, as well as including BO baseline, and additional benchmarking tasks. Additionally, we have corrected typos and updated the writing in the paper using many of your suggestions. Please let us know if this addresses your concerns.
>
> > "The current notation is often confusing and even ambiguous at times... I understand that notation preference is a very subjective thing so the authors should feel free to opt for something different, but I do think the notation needs to be improved."
>
> We have clarified ambiguous notation regarding expectations, loop indices in the pseudocode, and functions in the updated paper according to your suggestions. X and Y now always refer to the input and output variables, each expectation is now explicitly defined with the variable it is taken over, and summations are defined with the domain.
>
> > "Were comparisons with a simple approaches Bayesian optimization tried, e.g., Gaussian process?"
>
> We have included an additional Bayesian optimization baseline based on maximizing expected improvement on a Gaussian process posterior. We use an RBF kernel for the Superconductor task, and a dot-product kernel for the discrete GFP/Molecule tasks.
>
> > "The "thermometer encoding" of the output seems like an odd choice, especially given how it is done here...  the o_int_ks are parameters for logistic distributions and the mean is optimized through the unnormalized probs by optimizing each o_int_k directly. Would this be a fair description of the approach? "
>
> Your description is accurate - therefore we justified optimizing o_int_k directly by making a monotonicity argument, and optimizing the o_int_k simplifies the algorithm. However, we don’t consider this an “odd” choice since similar ideas have been applied previously - for example, the “discretized logistic” is used in works such as PixelCNN to convert a single scalar value into a categorical distribution.
>
> We have included an additional ablation study regarding the architecture choice in Appendix A.3.2, comparing the “thermometer” approach to the perhaps more straightforward method of training a model to predict discretized values directly. The results overall favor the thermometer architecture choice in performance and reliability. However, note that this is largely an architectural implementation detail -- a discretization design would be nicer, it's often the case that getting good results with deep neural networks requires careful design decisions, and we wanted to describe our decisions in detail for the sake of reproducibility.
>
> > "I don't think I understand what makes the appendix results an ablation study. From what I understand, these results only compare with the case where there is no learning of the model parameters. What are the models initialized to? Where does the data come into play?"
>
> The purpose of this ablation study was to measure the effect of using NML during optimization compared to optimizing a forward model directly.The models are initialized by pretraining on the dataset. Therefore, the difference between the two results is due to additional NML training. We have clarified this point in the appendix.
>
> > "How can I know that the method isn't just exploiting some quirk in the learned models used to evaluate it while some of the other methods avoid doing this?"
>
> In order to provide a “grounded” result, we have included additional results in the robotics domain from the Design-Bench benchmark. The designs in these tasks are evaluated in the MuJoCo simulator, and therefore high scores can be trusted to correspond to valid outputs. We find that the similar trends still hold true on these tasks - NEMO scores consistently high on each task.
>
> Unfortunately, it is difficult to gauge the validity of results on the material and molecule design tasks without synthesizing the outputs in real life. As discussed in Brookes et. al. 2019, the model class of the ground truth is a random forest and therefore belongs to a very different model class from the estimated function, and our model is trained on ground-truth values and not on the output of the random forest. Therefore, it is still a nontrivial problem to achieve high scores on these problems.
>
> > "What happens when doing more iterations on the log likelihood before updating x?"
>
> We have included a more detailed study in the Appendix A.3.3, comparing the ratio of NML model learning steps to optimization steps on x. When the model learning rate is small, taking additional steps appears to strictly improve the convergence speed of the method (as measured by overall algorithm steps). However, as somewhat expected, taking too many steps with a large learning rate can cause some instabilities.

---

> > ### Comment · AnonReviewer2 · 2020-11-18
> > **Response follow up**
> >
> > The detailed clarifications, the updated notation and additional results have made this a significantly stronger submission. I am pleased to see that authors appear to have invested a notable amount effort polishing this paper.
> >
> > ### Notation
> >
> > Notation is significantly improved and makes the paper much easier to read. There doesn't appear to be any noteworthy notation issues remaining.
> >
> > ### New experimental results
> >
> > * The addition of GP-BO complement well the previous results and help appreciate previous results.
> > * The addition of the mujoco results cover what I considered a major blind spot in the previous draft. These new results serve as a useful reference point which makes me more confident about the significance of the previous results. Furthermore, the mujoco results appears to support the claim that NEMO performs consistently well across tasks while other methods don't exhibit the same level of consistency. (The authors might want to render their best policy for hopper at some point. While it doesn't always payoff, the quirks of the mujoco simulator can be a great source of comical videos for talks, especially with a good optimization algorithm that can exploit them!)
> > * I appreciate that the authors have added more ablation results. As a general statement, I find results like these to be much more likely to inspire future research ideas or other minor conceptual breakthroughs, even if these results don't necessarily fit nicely into the "story".
> >
> > ### Thermometer
> >
> > First, I'd like to apologize for my choice of words which made my comment about thermometer encoding unconstructive at best. I should have caught that before submitting my initial review. What I should have said is that I was surprised by its introduction. I also believe I did a poor job expressing why so I will give it another go.
> >
> > Given the presentation up until that point in the paper, I expected that the conditional probability density would be approximated by a family of parametric probability density functions where the parameters would come from the output of some NN, and the denominator of (2) would then be approximated using a discrete set of points covering Y, i.e., a form of quadrature approximation of the integral. What tripped me up was just how similar the proposed approach was to what I expected but derived using the ideas of discretization and thermometer encoding.
> >
> > The authors should correct me if I'm wrong, but I believe that up until their assumption that $g(y) = y$ and other simplifications, the proposed method is equivalent to what I described when using a family of parametric logistic distribution with $\sigma = 1$. I think what I found "odd" was more the motivation/derivation of the method than the resulting method itself.
> >
> > Also, while I appreciate the addition of the softmax results, it was not my intention to suggest approximating that probability density with a softmax categorical distribution. I think the overall approach proposed by the authors is good and thought provoking. I appreciate papers that make you wonder what various extensions or generalization would look like and this was the case for me here.
> >
> > If my claim that the proposed method is equivalent (or almost) to using a parametric probability density and taking a quadrature approximation of the integral, the authors might want to consider using those concepts to introduce their method. I believe it would make the presentation more intuitive for many readers as well as allow the use of theory and ideas from quadrature methods in follow up work.
> >
> > To be clear, I'm writing this simply because I think it could improve this paper (provided I didn't misunderstand something, of course). The authors should feel free to opt for keeping the presentation as is. I consider the thermometer motivation to be sufficient and I don't believe changing it is necessary for publication.
> >
> > ### Remaining questions
> >
> > * How do the run times compare?
> > * Why were some of the Design-bench benchmark omitted? With the new results, I don't think the authors need to add them but I think the question is still relevant.
> >
> > ### Misc comments and typo
> >
> > * Algorithm 2, "for $x_t^m$ in $1 ... M$", should this be 'for $m$ in $1 ... M$'? Given the other changes to the notation, I suspect this might be an accidental omission.

---

> > > ### Author Response · Authors · 2020-11-19
> > > **Architecture presentation updated**
> > >
> > > On the thermometer encoding:
> > >
> > > Thank you for the clarifications. The quadrature + logistic distribution interpretation does seem to be a cleaner way to interpret the method, and we’ve updated Section 4.2 to use this interpretation rather than the original thermometer encoding. The implementation of the method itself remains unchanged. This should better motivate the use of quantization as an approximation scheme to computing a full integral, and as you mentioned, shed some light into potential future ways to improve the method.
> > >
> > > > “How do the run times compare?”
> > >
> > > To provide some concrete runtime numbers, here are runtimes we obtained on AWS c5.large instances for the Superconductor task.
> > >
> > > Optimizing a forward ensemble of 40 networks directly took on average 0.11s / gradient step on x. Adding 5 steps of NML model optimization (this was the setting used in our results reported in Table 1 & 2) brings this cost up to 0.35s / gradient step. So, when using equivalent neural network architectures and ensemble sizes, NEMO takes roughly 3x longer due to the cost of additional model training per iteration. In total, we ran our experiments for 50k gradient steps, which took a bit under 5 hours for NEMO on the Superconductor task, although 50k steps was more than the number of steps we needed to reach convergence.
> > >
> > > > “Why were some of the Design-bench benchmark omitted? With the new results, I don't think the authors need to add them but I think the question is still relevant.”
> > >
> > > With the inclusion of the robotics tasks, all 6 tasks in the Design-bench benchmark are now included in Table 1 & 2.
> > >
> > > Overall, thank you for the numerous suggestions. We think the paper has been greatly improved by your (and other reviewer’s) feedback.

---

> > > > ### Comment · AnonReviewer2 · 2020-11-19
> > > > **A happy reviewer**
> > > >
> > > > The new presentation for the discretization looks good! Since new results showing # NLM steps vs performance seem to suggest that better approximations improve performance, I wonder how big of an effect a more powerful numerical integral approximation scheme, e.g., a fixed Bayesian quadrature approach, would have. Maybe the authors are equally curious to know and my curiosity will be satisfied in follow up work :)
> > > >
> > > > ### How do the run times compare?
> > > >
> > > > Those are a bit better than I expected. Given the setting, runtime isn't all that important unless it starts to get prohibitively expensive but I wanted to do my due diligence and make sure nothing was swept under rug. It might be useful to add a few words with describing the "order of" what to expect in case a reader has the same concern, but that is far from necessary. I'm sure we can think of small things to add/change until the end of time and I don't want the authors to think they need to indulge me at this point! I think the authors have already done enough.
> > > >
> > > > ### Design-bench benchmark ~omitted~
> > > >
> > > > Ugh, I can't believe I missed that! That makes my comment quite silly...
> > > >
> > > > ### Closing thoughts
> > > >
> > > > I am quite happy with how the paper has progressed. The authors have addressed all my concerns and have considerably improved the overall quality of this paper. At this point, I believe this paper is ready for publication and I consider it clearly above the novelty and quality threshold for acceptance at ICLR. I will wait until later in the reviewer discussion phase before updating my review but I think it is reasonable for the authors to expect a notably improved score from me after my revision.

---

### Decision · Program_Chairs · 2021-01-07
**Final Decision**

**Decision:**

Accept (Poster)

**Comment:**

This work proposes a model-based optimization using an approximated normalized maximum likelihood (NML). It is an interesting idea and has the advantage of scaling to large datasets. The reviewers are generally positive and are satisfied with authors' response.